# 2-Deoxy-d-Glucose and Its Analogs: From Diagnostic to Therapeutic Agents

**DOI:** 10.3390/ijms21010234

**Published:** 2019-12-29

**Authors:** B. Pajak, E. Siwiak, M. Sołtyka, A. Priebe, R. Zieliński, I. Fokt, M. Ziemniak, A. Jaśkiewicz, R. Borowski, T. Domoradzki, W. Priebe

**Affiliations:** 1Independent Laboratory of Genetic and Molecular Biology, Kaczkowski Military Institute of Hygiene and Epidemiology, Kozielska 4, 01-163 Warsaw, Poland; ewelinasiwiak1@op.pl (E.S.); maja.soltyka@gmail.com (M.S.); ancpatrin@gmail.com (A.J.); radoslaw.borowski@wpdpharmaceuticals.com (R.B.); domoradzki.tomasz@gmail.com (T.D.); 2Texas Oncology, Division of Gynecologic Oncology, 910 E. Houston Street, Tyler, TX 75702, USA; aanyy@mac.com; 3Department of Experimental Therapeutics, The University of Texas MD Anderson Cancer Center, 1901 East Rd., Houston, TX 77054, USA; rziel77@gmail.com (R.Z.); ifokt@mdanderson.org (I.F.); 4Biological and Chemical Research Centre, Department of Chemistry, University of Warsaw, 02-089 Warsaw, Poland; mziemniak@chem.uw.edu.pl

**Keywords:** 2-deoxy-d-glucose, 2-DG analogs, glioblastoma, anticancer therapy

## Abstract

The ability of 2-deoxy-d-glucose (2-DG) to interfere with d-glucose metabolism demonstrates that nutrient and energy deprivation is an efficient tool to suppress cancer cell growth and survival. Acting as a d-glucose mimic, 2-DG inhibits glycolysis due to formation and intracellular accumulation of 2-deoxy-d-glucose-6-phosphate (2-DG6P), inhibiting the function of hexokinase and glucose-6-phosphate isomerase, and inducing cell death. In addition to glycolysis inhibition, other molecular processes are also affected by 2-DG. Attempts to improve 2-DG’s drug-like properties, its role as a potential adjuvant for other chemotherapeutics, and novel 2-DG analogs as promising new anticancer agents are discussed in this review.

## 1. Introduction

Cancer cells adopt different strategies allowing them to proliferate and invade other tissues, including avoidance of apoptosis, lack of sensitivity to growth inhibitors, independence from growth signals, unlimited replication potential, angiogenesis, immune escape, genetic instability, induction of chronic inflammation, and finally changes in cell metabolism [1,2]. Nobel Prize laureate Otto Warburg showed that cancer cells preferentially utilize glycolysis, even in the presence of oxygen, a process termed aerobic glycolysis or the Warburg effect [3]. Additionally, poorly vascularized regions within solid tumors are inherently hypoxic, further favoring glycolysis as a preferential pathway of ATP production by cancer cells [4]. Given that the majority of malignant cells preferentially utilize glycolysis, an opportunity exists to exploit this Achilles heel to selectively eliminate cancer cells while leaving untransformed cells unaffected. Although the diagnostic potential of this strategy has been widely explored, no therapeutic applications have been achieved thus far.

### 1.1. Glucose Metabolism

Glucose is the most common source of cellular energy and a substrate for many biochemical processes. It is typically produced from ingested dietary carbohydrates but may also be created within the body using gluconeogenesis [5]. The hydrophilic nature of glucose requires specific glucose transporter proteins (GLUTs) to facilitate cellular uptake [6]. Higher glucose utilization by tumor cells requires overexpression of GLUT transporters to increase glucose uptake over 20–30 fold as compared to normal cells [7,8].

Once inside the cell, glucose enters a cycle of changes to release energy in the form of ATP. Normal cells with access to oxygen utilize glycolysis to metabolize glucose into two molecules of pyruvate and form two molecules of ATP. The pyruvate is further oxidized in the mitochondria to acetyl-CoA via the pyruvate dehydrogenase complex. Acetyl-CoA then enters the Krebs cycle, in which it is oxidized into 2 molecules of CO_2_. The electrons derived from this process are used to create three molecules of NADH and one molecule of FADH_2_. These electron carrier molecules are reoxidized through the oxidoreductive systems of the respiratory chain, which drives ATP formation from ADP and inorganic phosphate (P_i_) [9]. As a result of oxidative phosphorylation, 30 ATP molecules are generated from one molecule of glucose versus a net of 2 from glycolysis [9,10]. Oxygen is vitally important for this process as the final electron acceptor, allowing complete oxidation of glucose. In the case of insufficient oxygen concentrations, for example in skeletal muscle during periods of intense exertion, cells fall back on glycolysis, an ancient metabolic pathway evolved before the accumulation of significant atmospheric oxygen. Pyruvate, the end product of glycolysis, is reduced to lactate via lactic acid fermentation, cycling NADH back to NAD^+^ [11]. The comparison of glucose metabolic pathways is presented in Figure 1.

### 1.2. Aerobic Glycolysis in Cancer Cells

Rapid tumor growth and the physical properties of solid tumors quickly outstrip the diffusion capacity of oxygen. To overcome this, tumors induce new blood vessel formation via angiogenesis. This leaky, unstable tumor neovasculature results in variable and unpredictable levels of hypoxia and subsequent lactic acidosis. To adapt to such conditions, cancer cells developed the ability to revert to anaerobic metabolism as their major source of ATP generation [12]. It is postulated that aerobic glycolysis is advantageous to tumor growth, increasing cell biomass, generating glucose-6-phosphate for the pentose phosphate pathway, and providing precursors for fatty acid, amino acid and nucleic acid synthesis [10], which may account for the predominance of glycolysis as the primary pathway for glucose metabolism, even in the presence of oxygen.

In response to hypoxic conditions, tumor cells reprogram protooncogenes (e.g., cellular myelocytomatosis oncogene (c-Myc)), modify signaling pathways (e.g., Phosphoinositide 3-kinase (PI3K/Akt), and activate specific transcription factors (e.g., hypoxia-inducible factor 1 alpha, HIF-1α) [10]. The HIF-1α transcription plays a key role in reprogramming cancer cell metabolism [13]. Normally activated in periods of starvation, HIF1α regulates the transcription of (genes encoding) glucose transporters and glycolytic enzymes, augments mitochondrial respiration by increasing the expression of pyruvate dehydrogenase kinase 1, and induces mitochondrial autophagy [14,15]. Furthermore, HIF-1α regulates the balance between oxygen consumption and production of ATP and toxic reactive oxygen species (ROS) [16]. Reprogramming oxidative metabolism to aerobic glycolysis is a major strategy contributing to survival and proliferation of cancer cells under hypoxic conditions [17].

In both aerobic and anaerobic glycolysis, the glucose molecule enters the cell via GLUT glucose transporters, is phosphorylated by hexokinase to glucose 6-phosphate and then it is either metabolized by glucose-6-phosphate dehydrogenase in the pentose phosphate pathway (PPP) or it is converted to fructose-6-P (F-6-P) via phosphoglucose-isomerase (PGI) [18]. In the next step, phosphofructokinase (PFK) catalyzes F-6-P to fructose-1,6-biphosphate (F-1 6-BP). This phosphorylation is ATP-dependent and irreversible, and is allosterically inhibited by excess ATP, making it a major regulator of glycolysis. F-1 6-BP is transformed either to glyceraldehyde-3-P or to dihydroxyacetone phosphate, which is later utilized in phospholipid and triacyloglycerol biosynthesis [19]. As glycolysis moves forward, phosphophenol pyruvate (PEP) is catalyzed to pyruvate by pyruvate kinase (PK) [19,20]. In the humans and other mammals, there are four isoenzymes of PK: pyruvate kinase muscle isozyme M1/M2 (PKM1/PKM2), red blood cell PK (PKR), and liver-type PK (PKL), which differ in their primary structure, kinetic properties, and tissue-specific expression [21]. The PKM2 isoform is considered a “prototype”, as it occurs in fetal tissues and stem cells, but is also noted in proliferating cells, including cancer cells. [22]. In normal cells, PKL also serves as a regulatory enzyme in gluconeogenesis, a metabolic pathway in the liver that converts pyruvate, lactate, and other substances to glucose during periods of starvation. When PK is inactivated by phosphorylation (which occurs during cell starvation due to the action of glucagon), PEP cannot be converted to pyruvate, and is instead converted to glucose via gluconeogenesis and distributed to tissues in a state of starvation [22]. High PKM2 activity has also been observed in tumor cells, allowing for high throughput glucose processing, stimulating rapid tumor growth [23]. Lastly, under anaerobic conditions in normal cells and in the presence of O_2_ in cancer cells, lactate is generated from pyruvate by lactate dehydrogenase (LDH), which concurrently oxidizes NADH to NAD+, allowing glycolysis to continue [23]. Schematic representation of aerobic glycolysis, such as is found in cancer cells, is shown in Figure 2.

While not all cancer types show high dependence on anerobic glucose metabolism [24], pancreatic ductal adenocarcinoma (PDAC) and glioblastoma multiforme (GBM)—two of the most aggressive and lethal types of tumors—are completely dependent on glycolysis [25,26].

## 2. Glycolysis as a Target in Anticancer Therapy

In recent decades, research efforts have focused on targeting cancer cell metabolism. Tumor glucose metabolism is an especially promising target, since it strongly differs from glucose metabolism in normal cells [27].

Arsenate compounds, 2-deoxy-d-glucose (2-DG), and 3-bromopyruvate (3-BrPA) are known to abolish ATP generation [28,29,30]. Pentavalent arsenic compounds do not directly inhibit glycolysis but, due to their resemblance to inorganic phosphate, disrupt ATP production by causing arsenolysis in the glyceraldehyde-3-phosphate dehydrogenase reaction, blocking generation of 1,3-bisphosphoglycerate [29]. Their use is limited, however, due to their significant toxicity. On the other hand, 3-BrPA and 2-DG were shown to inhibit hexokinase II (HKII), which catalyzes the initial metabolic step in conversion of glucose, and efficiently suppress glycolysis, and thus could be promising anticancer agents [30,31,32]. However, Kunjithapatham et al. [33] showed that 3-BrPa does not cross the blood–brain barrier (BBB), which is no obstacle for 2-DG, hence 2-DG but not 3-BrPa appeared as a candidate for brain tumor drugs.

2-DG is a synthetic glucose analog in which the 2-hydroxyl group is replaced by hydrogen (Figure 3).

Similarly to d-glucose, 2-DG is transported across the BBB and quickly taken up into cells mainly by glucose transporters (facilitated diffusion), in particular GLUT1 and GLUT4, although active transport via SGLT transporters also occurs [34,35]. Once inside the cells, 2-DG is phosphorylated to 2-deoxy-d-glucose-6-phosphate (2-DG-6-P), a charged compound that is trapped inside the cell. However, because it is missing the 2-OH group, it is unable to undergo isomerization to fructose-6-P, leading to intracellular accumulation of 2-DG-6-P and inhibition of glycolysis and glucose metabolism. As 2-DG is relatively non-toxic and orally available, it is an attractive tool for potential anticancer therapy [36]. Its diagnostic potential has been widely explored, but therapeutic applications have yet to be achieved. It seems that its most promising application may be as a synergystic agent in combination with cytotoxic therapies.

## 3. 2-DG in Cancer Diagnostics

Due to the Warburg effect, cancer cells consume more glucose than most normal cells, driving the development of 2-DG radioisotope analogs as probes to detect transformed cells. Nowadays, positron emission tomography (PET), based on the accumulation of a radiolabeled glucose analog (commonly ^18^F-DG) in tumor cells as a consequence of their enhanced glucose uptake and dependence on glycolysis [37], is commonly used for cancer detection, staging, monitoring response to treatment, and detection of recurrence in several tumor types. Moreover, due to the ability of 2-DG to freely penetrate the BBB, it has been proposed for use in brain tumor diagnostics [38]. The use of 2-DG in PET imaging was initiated in 1969 by Louis Sokoloff, who began research with [^14^C]DG, to measure cerebral glucose utilization because it accumulated in the brain [39]. Furthermore, Sokoloff and colleagues synthesized 2-[^18^F]fluoro-2-deoxy-d-glucose (^18^F-DG or FDG), one of the most widely used radiotracers today [40]. Due to the greater electronic similarity of fluorine to a hydroxyl group (versus the hydrogen atom in 2-DG), ^18^F-DG is more reminiscent of glucose [41]. PET scanning after intravenous injection of ^18^F-DG illustrates the uptake and distribution of the radiopharmaceutical, reflecting glucose uptake and metabolism throughout the body [42]. ^18^F-DG is converted to ^18^F-DG-6-phosphate and accumulates in tumors or organs that intensively metabolize or excrete glucose, such as the brain, heart, kidneys, ureters, and bladder [43]. repetitive

## 4. Biological Activity of 2-DG in Cancer Cells

Rather than simply acting as a roadblock in glycolysis, 2-DG interacts with multiple cellular pathways and has a range of biologic effects, which are further explored below.

### 4.1. Glycolysis Inhibition

For transport into the cell, 2-DG competes with glucose and can competitively inhibit glucose transport. Oxygen deficiency, more common in the intratumoral environment, increases the expression of glucose transporters and glycolytic enzymes, which increases 2-DG uptake in cancer cells as compared to normal cells in an aerobic environment [31,44,45]. After entering the cell, 2-DG is phosphorylated by hexokinase II to 2-deoxy-d-glucose-6-phosphate (2-DG-6-P) but, unlike glucose, 2-DG-6-P cannot be further metabolized by phosphoglucose isomerase (PGI) to a 5 carbon ring [46]. This leads to the accumulation of 2-DG-6-P within the cell, allosteric and competitive inhibition of hexokinase, isomerase depletion of ATP, cell cycle arrest and inhibition of cell growth, and eventually, cell death. Therefore, the greater the amount of accumulated 2-DG-6-P, the greater the effect on glycolysis [47]. Inhibition of glycolysis is more effective under hypoxic conditions, because in normoxia, cancer cells can continue to produce ATP using alternative sources, such as fatty acids or amino acids [47]. The schematic comparison of glucose and 2-DG metabolism in cancer cells and the cellular effects of 2-DG are illustrated in Figure 4.

### 4.2. Autophagy Induction

The main consequence of glycolysis inhibition is ATP deficiency, altering the ATP/AMP ratio and activating AMP-activated protein kinase (AMPK) [48]. Active AMPK directly phosphorylates the TSC1 proteins in the mTOR kinase complex (mammalian target of rapamycin kinase, mTORC), leading to induction of autophagy [49]. Autophagy is a degenerative mechanism present in every living cell [50]. In general, autophagy (herein referred to macroautophagy) is a process of sequestration of intracellular entities within double-membraned vesicles, called autophagosomes, and their delivery to lysosomes for degradation [51]. In cancer cells, autophagy has a dual role in tumor development and survival. Autophagy enhances tumor growth in the early growth phase [52] and, in advanced tumors, increases cancer cell survival and promotes metastasis [53].

AMPK activation may also induce the expression of p53, a tumor suppressor protein [54]. Active p53 arrests the cell cycle at the G1 checkpoint, allowing repair of cell damage or targeting it for destruction via apoptosis through the Bcl-2 family of proteins [55]. Further, p53 promotes processes related to oxidative phosphorylation by activating cytochrome c oxidase, reducing the expression of GLUT proteins or stimulating the TP53-inducible glycolysis and apoptosis regulator (TIGAR) protein, which inhibits phosphofructokinase (PFK) activity [23].

The degradation products of the autophagosomal cargo, which includes sugars, nucleosides or nucleotides, amino acids, and fatty acids, can be transported back to the cytoplasm, presumably to re-enter cellular metabolism. However, sustained autophagic flux may lead to excessive self-degradation of cellular components essential for survival (such as mitochondria), triggering cell death [56]. Regardless of mTOR activation, autophagy can be a consequence of endoplasmic reticulum (ER) stress induction in response to glucose deprivation and decreased ATP levels [47,57]. Further, Shutt et al. [58] demonstrated that 2-DG-mediated glucose deprivation is also able to stimulate production of reactive oxygen species (ROS) and the autophagy process, which could be blocked by upregulation of cellular antioxidant potential using a thiol antioxidant, *N*-acetyl-l-cysteine or catalase. Molecular pathways of autophagy induction via 2-DG are summarized in Figure 5.

### 4.3. Apoptosis Induction

Prolonged autophagy and extensive self-degradation ultimately leads to apoptotic cell death [59]. It appears that potent 2-DG apoptosis induction is transmitted via various pathways (autophagy, extrinsic, and intrinsic), depending on the cell type. Munoz-Pinedo et al. [60] reported that ATP depletion sensitizes cells to TNF superfamily death receptor dependent extrinsic apoptosis. Zhang et al. [44] found that 2-DG treatment significantly downregulates antiapoptotic cellular FLICE-like inhibitory protein (cFLIP) and receptor-interacting protein kinase (RIP) protein expression, leading to increased TNF-related apoptosis inducing ligand (TRAIL)-induced cell death. Similarly, Xu et al. [61] found that 2-DG can sensitize cells to TRAIL-induced apoptosis, at least in part through suppressing c-Jun N-terminal kinase (JNK)-mediated cytoprotective autophagy processes. Further, several reports have demonstrated the ability of 2-DG to increase ROS generation, leading to cell death [58,62,63]. Additionally, 2-DG has been shown to block the pentose phosphate shunt [41]. Decreased levels of NADPH correlate with reduced glutathione levels, one of the major cellular antioxidants. Valera et al. also observed that in bladder cancer cells, 2-DG treatment modulates the Bcl-2/Bax protein ratio, driving apoptosis induction [64].

### 4.4. Protein N-Glycosylation

*N*-glycans are major components of many secreted or membrane-bound glycoproteins involved in a number of important functions, including protein stabilization, folding, intracellular interactions and signal transmission. They are also involved in mediation of cell–cell and cell–extracellular matrix interactions and cell adhesion [65]. All eukaryotic *N*-linked glycans are based on the common core pentasaccharide, Man_3_GlcNAc_2_, comprising three d-mannose (Man) and two β-d-*N*-acetylglucosamine (GlcNAc) molecules. Through a cotranslational process, the glycan glucose_3_-mannose_9_*-N*-acetylglucosamine_2_ (G_3_Man_9_GlcNAc_2_) is transferred from a lipid-linked oligosaccharide (LLO), G_3_Man_9_GlcNAc_2_-P-P-dolichol, to Asn-X-Ser/Thr motifs and processed by various glycosyltransferases, giving rise to the three main types of glycans: high-mannose, hybrid, and complex *N*-glycans [66,67]. Perturbation of LLO synthesis results in aberrant *N*-linked glycosylation and interferes with glycan-dependent folding and quality control processes of the ER [68]. d-mannose is closely related to d-glucose, differing only in the stereochemistry at the C-2 position. The opposite orientation of the hydroxyl groups at C-2 makes these sugars C-2 epimers. The chemical structures of d-glucose and d-mannose are shown in Figure 6.

Historically, 2-DG was prepared from d-glucose by the elimination of the hydroxyl group at C-2. However, elimination of the hydroxyl group at C-2 in the d-mannose molecule leads to the same 2-DG compound. Thus, 2-DG can interfere in the metabolism of both d-glucose and d-mannose, including inhibiting glycolysis, disrupting mannose-related metabolic pathways, and competing with d-mannose in protein *N*-glycosylation [47,68,69,70,71]. Khaitan et al. postulated [72] that the efficacy of glycolysis or *N*-glycosylation inhibition depends on the oxygen concentration [72]. In hypoxic conditions, 2-DG preferentially acts as a glycolysis inhibitor, whereas in the presence of oxygen, it tends to interrupt protein glycosylation, leading to ER stress [68]. In line with this is an observation by Xi et al. [41] that showed that 2-DG induces autophagy, increases ER stress, and diminishes ATP levels, while addition of exogenous mannose prevents autophagy activation and ER stress but does not influence ATP levels.

## 5. Preclinical and Clinical Studies of 2-DG in Anticancer Therapy

Due to its ability to inhibit glycolysis and ATP synthesis, disrupt *N*-glycosylation of proteins, decrease energy metabolism and NADPH levels, and interfere with cellular thiol metabolism, generating oxidative stress, 2-DG appears to be an efficient cytotoxic agent. Importantly, all of these effects are mostly observed in cancer cells, without significant effect on the viability of normal cells [73]. Moreover, a unique ability to affect cancer cells under hypoxic conditions, which often limits traditional cytotoxic agents, has made 2-DG a promising candidate not only for monotherapy, but also as a component of combination therapy with other commercially available drugs, bioactive compounds, and radiotherapy.

### 5.1. 2-DG and Cytotoxic Chemotherapeutics

2-DG has been explored/tested as an adjuvant agent for various groups of clinically used chemotherapeutic drugs in breast, prostate, ovarian, lung, glioma, and other cancer types.

In this testing, 2-DG has been shown to sensitize cancer cells to cytotoxic drugs to which they are resistant when used alone [64]. Valera et al. [64] found that a non-toxic 5 mM 2-DG concentration sensitizes bladder cancer cells to doxorubicin, cisplatin, and gemcitabine, but not mitomycin C or 5′ fluorouracil (5-FU). However, in a pancreatic tumor model, 2-DG was able to sensitize cells to 5-FU, indicating a cell-specific effect [73].

According to Maher et al. [74], 2-DG activity could be limited in highly glycolytic tumors, such as pancreatic cancer, due to the high transcriptional activity of hypoxia-inducible factor 1α (HIF-1α), leading to increased amounts of glycolytic enzymes (mainly HK), transporting large amounts of glucose into the cell. Thus, these highly hypoxic and/or glycolytic cancer cells require higher concentrations of 2-DG to compete with glucose and effectively block glycolysis [74]. In such cases, combined therapy with HIF-1α inhibitors, such as WP1066, could restore cell sensitivity to 2-DG glycolysis inhibition [75].

Overall, it is plausible that 2-DG should be considered a potential adjuvant agent that has a selective, cell-specific chemosensitizing effect, improving the efficacy of standard chemotherapeutics. It is important to again underline the safety of 2-DG. Various animal and clinical studies demonstrate that 2-DG is safe and relatively non-toxic in animals and humans [76]. The most common adverse events after a 63 mg/kg dose of 2-DG were not life threating, and included fatigue, sweating, dizziness, and nausea, mimicking the symptoms of hypoglycemia [77]. The available research studies on 2-DG in combination therapy in preclinical and clinical studies are summarized in Table 1.

### 5.2. 2-DG and Radiotherapy Co-Treatment

Exposure to ionizing radiation (IR) results in cell damage via production of reactive oxygen species, as well as direct damage to nucleic acids and proteins (oxidation and bond breakage) [78]. Despite significant improvement in therapeutic techniques, there remain tumors with intrinsic resistance to radiation. It is well known that one of the major mechanisms of radioresistance is hypoxic cellular environment and enhanced glycolysis [77,79]. The lack of oxygen leads to decreased production of reactive oxygen species, and consequently, reduced DNA damage with radiotherapy [80]. It has been proposed that exposure to agents that enhance oxidative stress may sensitize cells to the cytotoxic effects of IR.

Additionally, 2-DG has been shown to inhibit DNA repair in highly glycolytic cells, such as respiratory-deficient yeast mutants [81]. This prompted researchers to evaluate the potential of 2-DG to improve the efficacy of radiotherapy. Several studies have demonstrated that 2-DG added prior to or immediately after IR enhances radiation-induced cell death by modifying energy-dependent cellular processes, such as DNA damage repair, cell cycle check points, and apoptosis [31,82,83]. Moreover, Lin et al. [78] suggested that 2-DG radiosensitization results from its effect on thiol metabolism, leading to ROS-dependent cell death, which could be prevented by *N*-acetyl-l-cysteine (NAC) treatment, downregulated glutamate cysteine ligase activity, or overexpression of ROS scavenging enzymes (e.g., superoxide and hydrogen peroxide). The radiosensitizing effect of adjuvant 2-DG in radiotherapy has been demonstrated in breast, prostate, cervical, lung, and GBM cancers (Table 1) [31,79,84,85].

Despite the numerous preclinical and clinical studies cited above, the use of 2-DG in cancer treatment is still limited. Its rapid metabolism and short half-life (according to Hansen et al., after infusion of 50 mg/kg 2-DG, its plasma half-life was only 48 min [117]) make 2-DG a rather poor drug candidate. Moreover, 2-DG has to be used at relatively high concentrations (≥5 mmol/L) in order to compete with blood glucose [118]. Adverse effects of 2-DG administration in humans include fatigue, sweating, dizziness, and nausea, mimicking the symptoms of hypoglycemia [77]. To overcome the abovementioned problems and to improve 2-DG’s pharmacokinetics and its drug-like properties, novel analogs of 2-DG were synthesized, and potential prodrugs prepared and tested.

## 6. Novel 2-DG Analogs

### 6.1. 2-Halogen Substituted d-Glucose

Lampidis et al. [119] synthesized and compared the physiological properties of a group of 2-halogenated d-glucose analogs: 2-fluoro-2-deoxy-d-glucose (2-FG), 2-chloro-2-deoxy-d-glucose (2-CG) and 2-bromo-2-deoxy-d-glucose (2-BG) (Figure 7).

Since glucose derivatives such as 2-DG compete with d-glucose to bind to GLUTs and hexokinase [120], further studies were conducted in order to determine the effect of the substituted halogen on activity. Lampidis et al. [119] used in silico ligand docking and energy minimalization for the protein–ligand complexes to examine the molecular interactions and binding energies between hexokinase I (HKI) in a series of halogenated analogs of 2-DG, and correlated with cytotoxicity in cancer cells. It appears that there is a negative correlation between the size of the halogen substituent at C-2 position and drug activity. As halogen size increases (2-FG > 2-CG > 2-BG), the ability of the halogenated glucose analog to bind the HKI active site is reduced, leading to decreased production of the 6-*O*-phosphorylated intermediates critical to glycolysis inhibition, decreased growth inhibition, and decreased cytotoxicity.

### 6.2. Fluoro-Hexose Compounds

Among the halogenated analogs of 2-DG, 2-FG is closest in structure to glucose due to the conformational and energetic similarity of fluorine to the hydroxyl group in glucose as compared to other halogens and hydrogen. Molecular modeling data indicate that 2-FG-6P may be comparable to G-6P in allosteric hexokinase inhibition. Since 2-FG is a more efficient substrate for HKII than 2-DG, its cellular uptake leads to higher accumulation of the reaction product, which inhibits glycolysis more efficiently. Treatment with 2-FG causes two- to three-fold higher inhibition of glycolysis in hypoxic cells than 2-DG. Under normoxic conditions, 2-FG showed small or even no activity [68]. Based on the structural similarity of d-glucose and d-mannose and the small size of fluorine atoms it was postulated that 2-fluoro-d-mannose (2-FM) (Figure 8) may also have comparable activity against tumor cells. 

It was confirmed that 2-FM, similarly to 2-DG, competes with mannose for *N*-linked protein glycosylation [121,122,123]. However, 2-DG competes with mannose and incorporates into lipid-linked oligosaccharides (LLOs), whereas 2-FM does not. Thus, 2-FM is a less potent inhibitor of protein glycosylation than 2-DG [68]. According to in vitro toxicity studies in 1420 and SKBR3 breast cancer cell lines, the potential to inhibit glycoprotein synthesis by 2-DG analogs is in the following order: 2-DG > 2-FM > 2-FG [68,122]. Comparative tests of 2-FM treatment in normoxia and hypoxia in U87 GBM cells showed the ability to induce autophagic cell death under normoxia [123]. Transmission electron microscopy revealed the presence of autophagosomes in response to 5 mM 2-FM action in U87 cells [123]. Thus, 2-FM and 2-DG, but not 2-FG, are toxic to selected tumor cell lines growing under normoxic conditions due to interference with glycosylation. The abovementioned compounds have been found to be effective against non-small cell lung (NSCLC), breast (SKBR3), osteosarcoma, and glioma cell lines, even under normoxic conditions. Intriguingly, cell lines derived from related tissues—meaning non-transformed—were found to be resistant to these compounds in the presence of oxygen, indicating that the toxicity of mannose analogs may not be tissue-type-specific [68].

### 6.3. Acetates of 2-Deoxy Monosaccharides as Prodrugs

A highly promising novel group of compounds, the acetyl 2-DG analogs, have been developed in Dr. Waldemar Priebe’s laboratory. Among the tested derivatives, lead compound WP1122 (3,6-di-O-acetyl-2-deoxy-d-glucose) has been selected for further studies [124] (Figure 9).

WP1122 enters cells and, importantly, crosses the BBB by passive diffusion rather than relying upon a specific glucose transporter. It undergoes deacetylation by intracellular esterases, and is thereafter trapped inside the cell after phosphorylation at the C-6 hydroxyl group. The resultant 6-phospho-2-deoxyglucose acts as a competitive inhibitor of HK, blocking phosphorylation of d-glucose, and thereby inhibiting the glycolytic pathway [124]. The ability of WP1122 to target brain tissue makes it a serious candidate for the treatment of brain tumors, including GBM. It has been shown that whereas 2-DG is rapidly metabolized, the prodrug WP1122 releases 2-DG, increasing its half-life and increasing therapeutic effect. WP1122 demonstrates good oral bioavailability, resulting in a two-fold higher plasma concentration of 2-DG than that achieved via administration of 2-DG alone [124]. In vitro experiments confirm that WP1122 potently inhibits glycolysis in U87 cell lines as determined by real-time monitoring of the extracellular acidification rate (ECAR), resulting in 2–10 times more potent anticancer activity when compared to 2-DG (half maximal inhibitory concentration (IC_50_) range of 1–10 mM) in both hypoxic and normoxic conditions [124]. Moreover, WP1122 is well tolerated by mice in an orthotropic GBM model, even with prolonged exposure. Recently, Keith et al. [125] demonstrated significant, real-time inhibition of glycolysis by WP1122 in an orthotopic brain tumor model. The authors compared the pyruvate to lactate conversion rate after oral administration of WP1122 using hyperpolarized pyruvate magnetic resonance (MR) spectroscopy. WP1122 is currently licensed to Moleculin Inc. and is planned to enter clinical trials for GBM patients.

## 7. Perspectives

Over the last decade, we have seen amazing progress in research on 2-DG and its analogs. Importantly, most of their mechanisms of action, many of which could have useful applications in anticancer therapy, have been described. Clinical trials of 2-DG have demonstrated the challenges in its use in monotherapy due to poor drug-like characteristics [124], leading researchers to focus on improving bioavailability and achieving higher therapeutic concentrations. Novel 2-DG analogs such as WP1122 and others have renewed the old concept of glycolysis inhibition as an effective anticancer strategy. Combined with other potent cytotoxic agents, novel glucose analogs could synergistically eliminate cancer cells. Our future efforts are focused on the development of new combinations of anticancer agents coupled with novel 2-DG derivatives targeting pancreatic ductal adenocarcinoma and glioblastoma, both of which are in desperate need of novel approaches and therapeutic options and particularly suited to glycolysis inhibition due to their reliance on aerobic glycolysis.

## Figures and Tables

**Figure 1 ijms-21-00234-f001:**
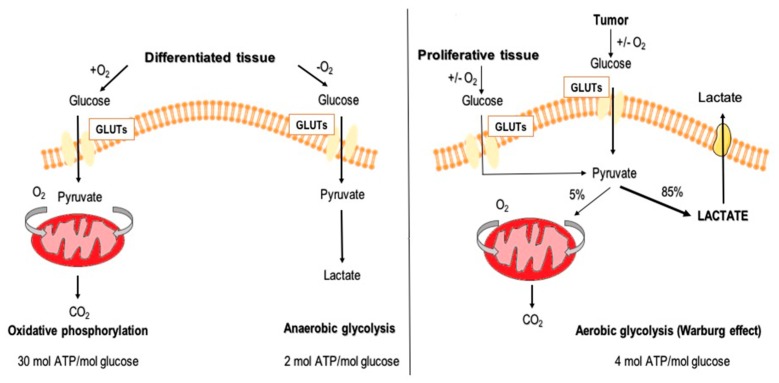
Schematic representation of glucose metabolic pathways: oxidative phosphorylation, anaerobic glycolysis, and aerobic glycolysis (glucose transporters, GLUTs).

**Figure 2 ijms-21-00234-f002:**
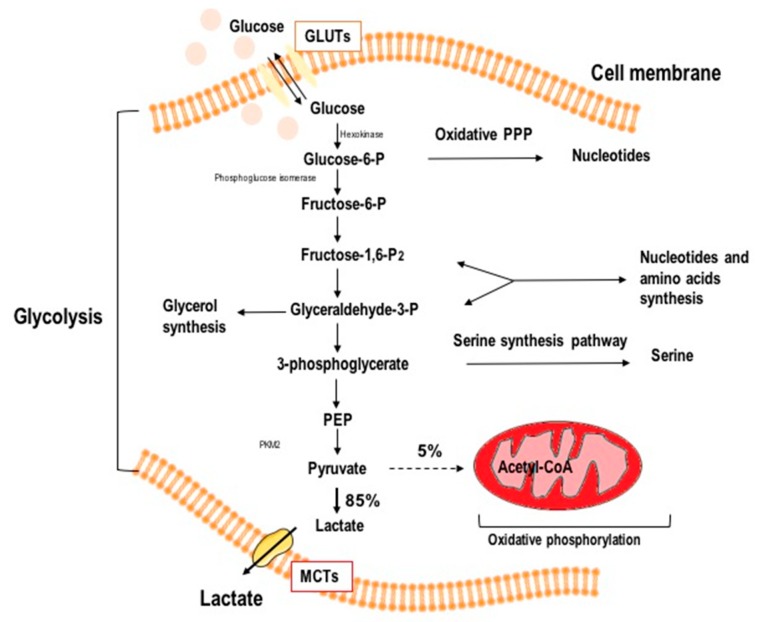
Schematic illustration of aerobic glycolysis in cancer cells (GLUTs—glucose transporters; MCTs—monocarboxylate transporters; PPP—pentose phosphate pathway; PEP—phosphophenol pyruvate; PKM2—pyruvate kinase isozyme M2). Pyruvate is preferentially shunted to lactate, resulting in increased lactate production. Oxidative metabolism persists at a low rate but is uncoupled from increased glycolysis.

**Figure 3 ijms-21-00234-f003:**
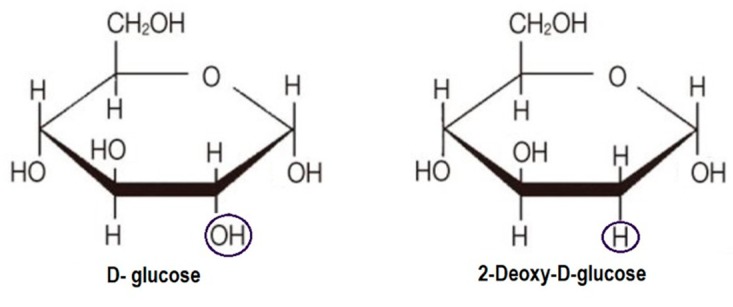
Molecular structure of d-glucose and 2-DG. The position of the hydroxyl group in 2-DG is indicated.

**Figure 4 ijms-21-00234-f004:**
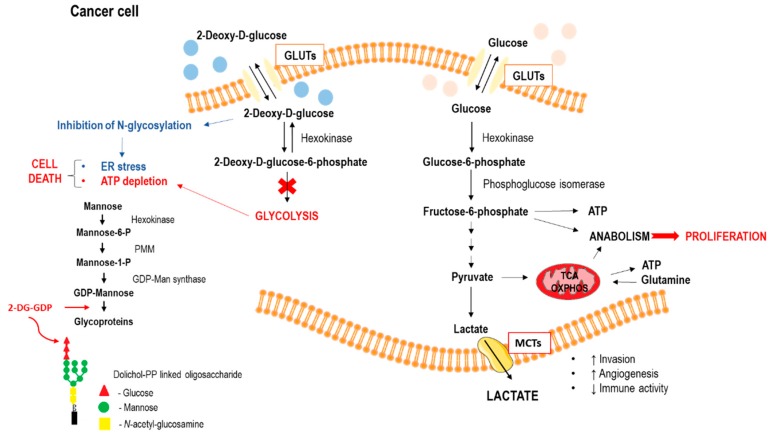
Overall scheme of 2-DG and glucose metabolism in cancer cell (GLUTs—glucose transporters; MCTs—monocarboxylate transporters; ER—endoplasmic reticulum; MT—mitochondrion; TCA—tricarboxcylic acid cycle; OXPHOS—oxidative phosphorylation; PMM—phosphomannosemutase).

**Figure 5 ijms-21-00234-f005:**
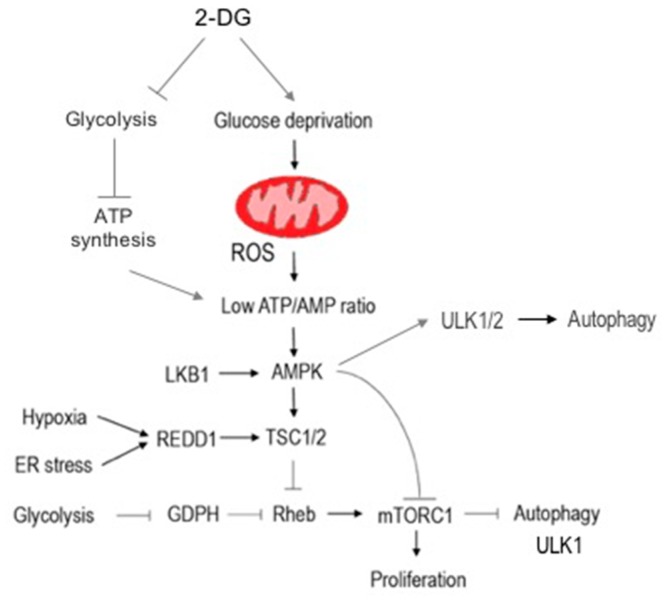
The 2-DG-induced autophagy pathway (Unc51 like autophagy activating kinase 1 (ULK1/2)—UNC-51-like kinase ½ (serine/threonine kinase homologous to yeast Atg1); REDD1—regulated in development and DNA damage response 1; GDPH—glycerol-3-phosphate dehydrogenase; Rheb—Ras homolog enriched in brain; LKB1—liver kinase B1). The main consequence of glycolysis inhibition is ATP deficiency, which disrupts ATP/AMP ratio. As a result, AMP-activated protein kinase (AMPK) is activated. Active AMPK directly phosphorylates tuberous sclerosis (TSC1) proteins in mammalian target of rapamycin (mTOR) kinase complex leading to autophagy induction. On the other hand, 2-DG-mediated glucose deprivation stimulates reactive oxygen species (ROS) production in mitochondria, also leading to AMPK activation and autophagy stimulation.

**Figure 6 ijms-21-00234-f006:**
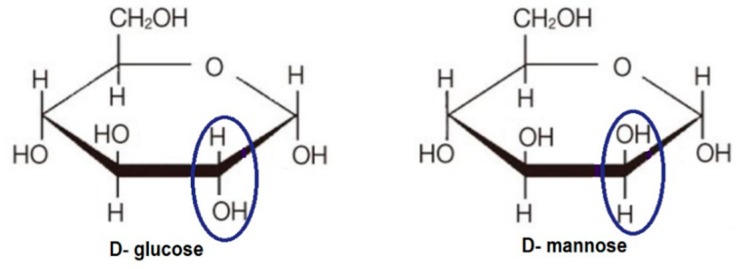
Molecular structures of d-glucose and d-mannose.

**Figure 7 ijms-21-00234-f007:**
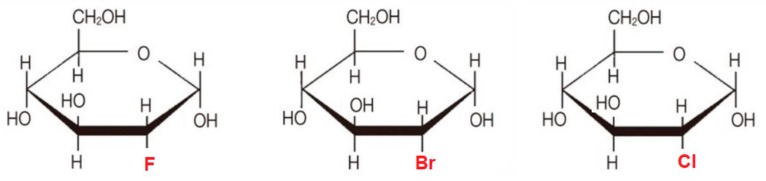
Chemical structures of (from the left) 2-fluoro-2-deoxy-d-glucose (2-FG), 2-bromo-2-deoxy-d-glucose (2-BG), and 2-chloro-2-deoxy-d-glucose (2-CG).

**Figure 8 ijms-21-00234-f008:**
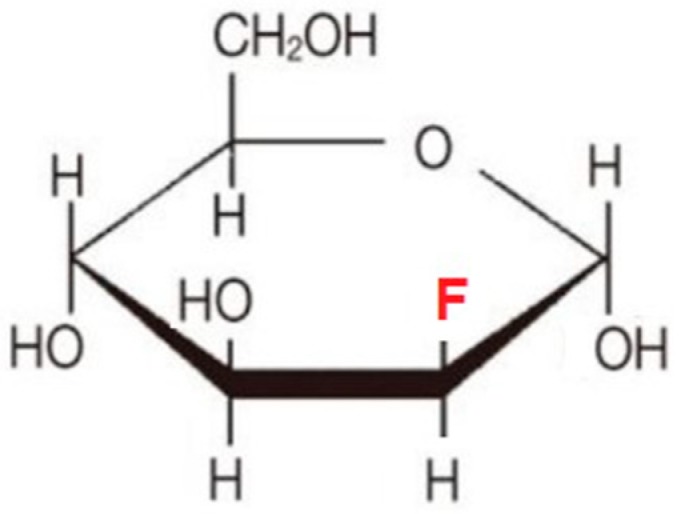
Chemical structure of 2-fluoro-mannose (2-FM).

**Figure 9 ijms-21-00234-f009:**
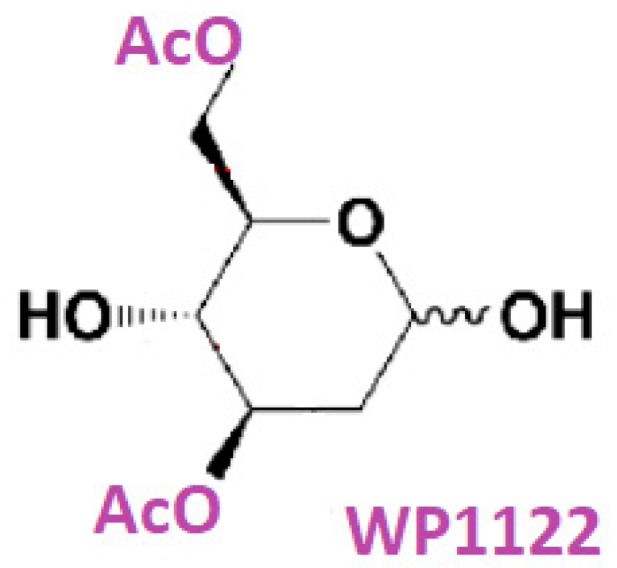
Chemical structure of 3,6-di-*O*-acetyl-2-deoxy-d-glucose (WP1122).

**Table 1 ijms-21-00234-t001:** Preclinical and clinical studies of combined anticancer therapy with 2-DG.

Combined Therapy with 2-DG	Cancer Type	References
**Preclinical Studies**		
Cisplatin	- head and neck carcinoma	[64,86,87]
	- GBM	
	- bladder cancer	
Metformin	- breast cancer	[88,89,90,91]
	- B cell lymphoma cells	
	- ovarian cancer	
	- GBM	
NCL-240	- melanoma	[92]
	- small lung carcinomas	
	- ovarian cancer	
	- breast cancer	
Doxorubicin	- papillary thyroid carcinoma	[64,93,94,95]
	- breast cancer	
	- bladder cancer	
Daunorubicin	- colon cancer	[96]
	- breast cancer	
Gemcitabine	- bladder cancer	[64]
Sorafenib	- papillary thyroid carcinoma	[93,97,98]
	- hepatocellular carcinoma	
Adriamycin	- osteosarcoma	[99]
	- non-small cell lung cancer	
Barasertib and Everolimus	- leukemia	[100]
Salirasib	- pancreatic cancer	[101]
Paclitaxel	- osteosarcoma	[99]
	- non-small cell lung cancer	
Berberine	- lung cancer	[102]
Fenofibrate (FF)	- breast cancer	[103]
	- melanoma	
	- osteosarcoma	
Resveratrol	- neuroblastoma	[104]
Mito-Q, Mito-CP, Dec-TPP+	- breast cancer	[105]
Methylprednisolone	- non-Hodgkin lymphoma	[106]
Alpha-tocopheryl succinate	- colon adenocarcinoma	[107]
	- cervical carcinoma	
	- lung adenocarcinoma	
Afatinib	- non-small cell lung cancer	[108]
Etoposide	- Ehrlich ascites tumor-bearing mice	[109]
Oligomycin	- small cell lung cancer	[110,111]
	- GBM	
Bevacizumab	- GBM	[112]
5′-Fluorouracil	- pancreatic cancer	[73]
Trastuzumab	- breast cancer	[113]
Ferulic acid with irradiation	- non-small cell lung carcinoma	[114]
Radiotherapy	- breast cancer	[31,78,85]
	- prostate cancer	
	- cervical cancer	
Virotherapy (avian Newcastle disease virus (NDV))	- breast cancer	[115]
**Clinical Studies**		
Docetaxel	- breast cancer,	[77]
	- lung cancer	
	- head and neck cancers	
Dehydroepiandrosterone (DHEA)	- breast cancer	[116]
	- prostate cancer	
Radiotherapy	- GBM	[84]

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
