# Peer review of "2-Deoxy-d-Glucose and Its Analogs: From Diagnostic to Therapeutic Agents"

_ijms, 2019, doi:10.3390/ijms21010234_

Round 1
Reviewer 1 Report
In this manuscript, Siwiak et al. reviewed the present research progress of 2-Deoxy-D-glucose and its analogs especially for their roles as diagnostic agents and adjuvant for anticancer therapy. The topic is not novel, but it can be considered for publication after revision.
Line 83 Figure 1. The name of “aerobic glycolysis” was not shown in the schematic diagram. The diagram on the right is mislabeled.
Line 125 Figure 2. What do dotted and solid lines mean? Are there any differences between the two?
Line 141, only its radioisotope analogues were possible cancer probes, but not the 2-DG glucose itself.
Line 286, it is better to illuminate “the poor drug-like properties” of 2 DG in detail.
Line 322-323, is the decreased ROS production the main cause of radiation resistance? Please refer to the following literature: Nat Rev Cancer 2015; 15(7):409-425.
Line 337-342, what is the difference between 2-fluoro-2-deoxy-D-glucose (2-FG) and 2-[18F] fluoro-2-deoxy-D-glucose (18F-DG) (line 149)? In addition, is the main purpose of synthesizing 2-Halogen Substituted D-Glucose is to extend the half-life? Furthermore, the author didn’t mention the improvement of the metabolism and the half-life about the majority of the 2-DG derivatives.
Author Response
Thank you very much for the comments of the reviewers, who very accurately indicated the weak points of our paper. All comments were taken into account and the changes visibly enriched the content of the paper. We hope that the reviewers will be satisfied with the current form of the manuscript.
After receiving the review, we consulted the content of our publication with our colleagues from USA, engaged in of the project focused on new 2-DG derivatives – Professor Waldemar Priebe and Anna Priebe, RafaÅ‚ Zielinski and Izabela Fokt. Described analogues and described anticancer strategy are created in Prof. Priebe’s laboratory. In addition to taking into account the comments of the reviewers, we decided to significantly improve the manuscript and change some paragraphs to make them more legible. W. Priebe, R. Zielinski, A. Priebe and I. Fokt significantly contributed in manuscript corrections thus we all agreed to change the list of co-authors. All previous authors agreed to include them in the paper..
Below are answers to reviewers' comments.
Review 1
Comments and Suggestions for Authors
In this manuscript, Siwiak et al. reviewed the present research progress of 2-Deoxy-D-glucose and its analogs especially for their roles as diagnostic agents and adjuvant for anticancer therapy. The topic is not novel, but it can be considered for publication after revision.
Line 83 Figure 1. The name of “aerobic glycolysis” was not shown in the schematic diagram. The diagram on the right is mislabeled.
Figure has been corrected.
Line 125 Figure 2. What do dotted and solid lines mean? Are there any differences between the two?
It was our mistake. The figure legend has been corrected.
Line 141, only its radioisotope analogues were possible cancer probes, but not the 2-DG glucose itself.
The statement has been corrected.
Line 286, it is better to illuminate “the poor drug-like properties” of 2 DG in detail.
2-DG properties have been described (page 11).
Line 322-323, is the decreased ROS production the main cause of radiation resistance? Please refer to the following literature: Nat Rev Cancer 2015; 15(7):409-425.
ROS production is obviously not the main cause of radiation resistance and the sentence has been corrected. The suggested reference has been added.
Line 337-342, what is the difference between 2-fluoro-2-deoxy-D-glucose (2-FG) and 2-[18F] fluoro-2-deoxy-D-glucose (18F-DG) (line 149)? In addition, is the main purpose of synthesizing 2-Halogen Substituted D-Glucose is to extend the half-life? Furthermore, the author didn’t mention the improvement of the metabolism and the half-life about the majority of the 2-DG derivatives.
2-fluoro-D-glucose (2-FG) and 2-[18F] fluoro-2-deoxy-D-glucose (18F-DG) are the same 2-DG analogues. The only difference is the presence of radioisotope, useful for diagnostic purposes. The information about improvement of the metabolism and the half-life of 2-DG derivatives, has been added in the text.
Submission Date
22 November 2019
Date of this review
01 Dec 2019 19:36:14
Review 2
Comments and Suggestions for Authors
A review on 2-DG is very welcome. First, the authors present glucose metabolism and the importance of glycolysis in cancer cells. Then, they summarize the importance of 2-DG in cancer diagnostic. They further develop the biological activities of 2-DG in cancer cells and focus on the use of 2-DG in combined therapy. They complete the review with interesting perspectives on novel 2-DG analogues. The review is of interest pointing out the key issues on 2-DG. The figures have to be improved and completed with full legends.
Some remarks are listed below:
1)The first paragraphs of the review are the most confused with heavy phrasing and spelling mistakes.
For instance,
Depravation instead of deprivation several times in the abstract and the main text.
Lane 3-35 I do not understand the sentence. What is “particular organism”?, repetition “type”
Lane 41: resistant instead of resistance
Lane 44: I do not understand the sentence
Lane 45: “disturbed balance between proliferation, maturation and differentiation of tumor cells cause…” not clear for me. What is maturation?
Lane 54 …is also the preferential
Lane 59 “typical cells” What does it mean?
Lane 64 proteins that instead of there?
Which kind of protein linked to glucose metabolism are specifically expressed in this context?
Lane 158 “is” repetition
Lane 164: suppress hexokinase II activity, an…
Lane 169: 3-bromopyruvate has to be modified
And more has to be checked…
The manuscript has been carefully checked and all mentioned-above mistakes and more have been corrected. Furthermore, the text has been re-written to be clearer.
2) Lane 179: the objective of the review appears at the end of the third paragraph!?
- The objective of the review has been moved to the first paragraph.
3) Lane 198 Are you sure the inhibition of HKII is a competitive one? Several papers mention the blocage of both HKII and PGI by 2-DG. Phosphoglucoisomerase and hexokinase are respectively competitively and non-competitively inhibited by 2-DG-6-phosphate.
- Chen W, Gueron M. The inhibition of bovine heart hexokinase by 2-deoxy-D-glucose-
6-phosphate: characterization by 31P NMR and metabolic implications. Biochimie 1992;74(9-10):867-73.
- Wick AN, Drury DR, Nakada HI, Wolfe JB. Localization of the primary metabolic block produced by 2-deoxyglucose. The Journal of biological chemistry 1957;224(2):963-9.
- We agree with the reviewr that 2-DG is a non-competitive inhibitor of HKII. The publications were verified and added in the references. The statement has been corrected.
4) Is there a competition between glucose and 2-DG for cell entry?
- Yes, 2-DG and glucose compete with each other for GLUTs, thus high concentration of 2-DG has to be used to achieve its therapeutic potential. It was explained in the text (page 6).
5) 1.2 Aerobic Glycolysis in Cancer Cells
This paragraph deals a lot with hypoxic conditions, steps of glycolysis and I do not see the focus on aerobic glycolysis.
- The paragraph has been corrected.
6) 4.4 Proteins N-glycosylation
The mode of action of 2-DG on the N-linked glycosylation should be explained with more details. This remark is in accordance with the comments on figure 5.
The last part of the § starting from “Overall, it was shown…” sound off topic, not dealing with N-glycosylation.
The paragraph has been corrected according to the Reviewer’s suggestion.
7) Figure 5 : inhibition of N-glycosylation should be schematized with more details. What means the dashed lines?
- Figure and figure legend have been corrected.
8) Figure 1: 85% and 5% are not cited in the text and associated with no reference. Is it anaerobic or aerobic glycolysis associated with Warburg Effect?
- Warburg Effect is of course associated with aerobic glycolysis. Suggested reference has been added in the text [line 115-116].
9) Figure 2 is not optimal. We do not see clearly where is the adaptation of the cell and what is specific to the aerobic glycolysis.
- The Figure 2 has been improved.
10) Figure 6 : 2-DG should appear in the figure. This figure looks quite basic. AMPK phosphorylation is not presented for example. Why ROS?
- Figure 6 has been corrected and more detailed description has been added in the text. It was shown by
by Shutt et al., that 2-DG-mediated glucose deprivation stimulates reactive oxygen species (ROS) production and treatment with the thiol antioxidant, N-acetyl-L-cysteine or catalase, protected cells from 2-DG—induced cell death.
Shutt, D.C.; O'Dorisio, M.S.; Aykin-Burns, N.; Spitz, D.R. 2-deoxy-D-glucose induces oxidative stress and cell killing in human neuroblastoma cells. Cancer. Biol. Ther. 2010, 9(11), 853-61.
Furthermore, it was shown that in general, glucose deprivation causes oxidative stress and stimulates autophagy via AMPK activation [Marambio et al. Biochim Biophys Acta 2010; 1802: 509-18].
11) Are there data estimating the relative contribution of glycolysis inhibition and N-linked glycosylation interference in the anticancer effects of 2-DG ?
- Up today, we did not find any studies estimating the contribution of these two processes in targeted cells. There were hypothesis that in hypoxic conditions, 2-DG preferentially inhibits glycolysis, whereas in normoxic conditions N-glycosylation process is privileged. It was shown that in in vitro studies, addition of mannose is able to decrease the cytotoxic effect of 2-DG. However, in animal or clinical model, the contribution of particular pathway could be modulated by several factors. More importantly, the overall cytotoxic effect of 2-DG has always been a priority.
12) What are the side effects of 2-DG on patients? How about 2-DG effects on organs depending almost exclusively on glucose such as the brain?
- According to Raez et al. studies [84], in clinical studies the most common adverse events after 63 mg/kg 2-DG dose were fatigue, sweating, dizziness and nausea mimicking the hypoglycemic symptoms, typical for limiting access of glucose to the brain. The most significant adverse effects noted at 63-88 mg/kg doses were reversible hyperglycemia (100%), gastrointestinal bleeding (6%) and reversible grade 3 QTc prolongation (22%). It was also explained in the text [lines 407-411].
13) Another original co-treatment 2-DG/virotherapy : 2-Deoxyglucose and Newcastle Disease Virus Synergize to Kill Breast Cancer Cells by Inhibition of Glycolysis Pathway Through Glyceraldehyde3-Phosphate Downregulation. Front. Mol. Biosci., 27 September 2019 | https://doi.org/10.3389/fmolb.2019.00090
- Thank you. The suggested co-treatment has been added in the Table and in the Reference section.
14) The figures have to be improved and completed with full legends.
- Figures have been corrected and figure legends were improved.
Reviewer 2 Report
A review on 2-DG is very welcome. First, the authors present glucose metabolism and the importance of glycolysis in cancer cells. Then, they summarize the importance of 2-DG in cancer diagnostic. They further develop the biological activities of 2-DG in cancer cells and focus on the use of 2-DG in combined therapy. They complete the review with interesting perspectives on novel 2-DG analogues. The review is of interest pointing out the key issues on 2-DG. The figures have to be improved and completed with full legends.
Some remarks are listed below :
1)The first paragraphs of the review are the most confused with heavy phrasing and spelling mistakes.
For instance,
Depravation instead of deprivation several times in the abstract and the main text.
Lane 3-35 I do not understand the sentence. What is “particular organism”?, repetition “type”
Lane 41 : resistant instead of resistance
Lane 44 : I do not understand the sentence
Lane 45 : “disturbed balance between proliferation, maturation and differentiation of tumor cells cause…” not clear for me. What is maturation?
Lane 54 …is also the preferential
Lane 59 “typical cells” What does it mean?
Lane 64 proteins that instead of there?
Which kind of protein linked to glucose metabolism are specifically expressed in this context?
Lane 158 “is” repetition
Lane 164 : suppress hexokinase II activity, an…
Lane 169 : 3-bromopyruvate has to be modified
And more has to be checked…
2) Lane 179 : the objective of the review appears at the end of the third paragraph!?
3) Lane 198 Are you sure the inhibition of HKII is a competitive one? Several papers mention the blocage of both HKII and PGI by 2-DG. Phosphoglucoisomerase and hexokinase are respectively competitively and non-competitively inhibited by 2-DG-6-phosphate.
- Chen W, Gueron M. The inhibition of bovine heart hexokinase by 2-deoxy-D-glucose-
6-phosphate: characterization by 31P NMR and metabolic implications. Biochimie 1992;74(9-10):867-73.
- Wick AN, Drury DR, Nakada HI, Wolfe JB. Localization of the primary metabolic block produced by 2-deoxyglucose. The Journal of biological chemistry 1957;224(2):963-9.
4) Is there a competition between glucose and 2-DG for cell entry ?
5) 1.2 Aerobic Glycolysis in Cancer Cells
This paragraph deals a lot with hypoxic conditions, steps of glycolysis and I do not see the focus on aerobic glycolysis.
6) 4.4 Proteins N-glycosylation
The mode of action of 2-DG on the N-linked glycosylation should be explained with more details. This remark is in accordance with the comments on figure 5.
The last part of the § starting from “Overall, it was shown…” sound off topic, not dealing with N-glycosylation.
7) Figure 5 : inhibition of N-glycosylation should be schematized with more details. What means the dashed lines?
8) Figure 1 : 85% and 5% are not cited in the text and associated with no reference. Is it anaerobic or aerobic glycolysis associated with Warburg Effect?
9) Figure 2 is not optimal. We do not see clearly where is the adaptation of the cell and what is specific to the aerobic glycolysis.
10) Figure 6 : 2-DG should appear in the figure. This figure looks quite basic. AMPK phosphorylation is not presented for example. Why ROS?
11) Are there data estimating the relative contribution of glycolysis inhibition and N-linked glycosylation interference in the anticancer effects of 2-DG ?
12) What are the side effects of 2-DG on patients? How about 2-DG effects on organs depending almost exclusively on glucose such as the brain?
13) Another original co-treatment 2-DG/virotherapy : 2-Deoxyglucose and Newcastle Disease Virus Synergize to Kill Breast Cancer Cells by Inhibition of Glycolysis Pathway Through Glyceraldehyde3-Phosphate Downregulation. Front. Mol. Biosci., 27 September 2019 | https://doi.org/10.3389/fmolb.2019.00090
14) The figures have to be improved and completed with full legends.
Author Response

(The authors gave the same response as above.)
